# Support for Mothers, Fathers, or Guardians of Transgender Children and Adolescents: A Systematic Review on the Dynamics of Secondary Social Networks

**DOI:** 10.3390/ijerph19148652

**Published:** 2022-07-16

**Authors:** Paula D. Abreu, Rubia L. P. Andrade, Israel L. S. Maza, Mariana G. B. F. Faria, Ana B. M. Valença, Ednaldo C. Araújo, Pedro F. Palha, Ricardo A. Arcêncio, Ione C. Pinto, Jaqueline G. A. Ballestero, Sandra A. Almeida, Jordana A. Nogueira, Aline A. Monroe

**Affiliations:** 1Ribeirao Preto College of Nursing, University of Sao Paulo, Ribeirao Preto 14040-902, Brazil; pauladdabreu@usp.br (P.D.A.); israelucas2008@gmail.com (I.L.S.M.); rimagod@usp.br (M.G.B.F.F.); palha@eerp.usp.br (P.F.P.); ricardo@eerp.usp.br (R.A.A.); ionecarv@eerp.usp.br (I.C.P.); jaqueco@gmail.com (J.G.A.B.); amonroe@eerp.usp.br (A.A.M.); 2Nursing Department, Federal University of Pernambuco, Recife 50670-901, Brazil; beatriz.valenca@ufpe.br (A.B.M.V.); ednaldo.araujo@ufpe.br (E.C.A.); 3Clinical Nursing Department, Federal University of Paraiba, Joao Pessoa 58051-900, Brazil; sandraalmeida124@gmail.com (S.A.A.); jalnogueira31@gmail.com (J.A.N.)

**Keywords:** mothers, parents, transgender persons, social support, social networking

## Abstract

Mothers’, fathers’, or guardians’ support for disclosures of diverse gender identity has significant relationships with decreased suicidality for transgender children and adolescents. They play an essential role in facing transphobia, protecting trans children, and strengthening the expression of their identity. These guardians need structural, emotional, and informative support; they need to be prepared to recognize and manage of their own feelings, as well as deal with the challenges that come with new social contexts of transphobia in schools, health institutions, and other community spaces. This study aimed to analyze the scientific evidence on the dynamics of secondary social networks to support mothers, fathers, or guardians of transgender children and adolescents. This is a systematic review of qualitative studies, guided by PRISMA guidelines. Controlled and free vocabularies were used to survey the primary studies in the following databases: EMBASE; Scopus; MEDLINE; Cumulative Index to Nursing and Allied Health Literature (CINAHL); PsycInfo; Latin American and Caribbean Literature in Health Sciences (LILACS); and Web of Science. A total of 28 articles made up the final sample of this review. Secondary social networks were described as fragile, characterized by conflicting and broken ties with healthcare services and professionals, isolation and unpreparedness from schools, and emotional and informational support from peer groups and some qualified healthcare professionals. The literature shows the potential of the dynamics of secondary social support networks; however, it presented the unpreparedness of professionals and institutional policies for welcoming transgender children and adolescents and their families, with the peer group being the main emotional and informative support network.

## 1. Introduction

Gender-diverse youth are significantly more likely to reveal their identity to the people in their lives, including their parents, than their cisgender and heterosexual counterparts. In comparison, parents of gender-diverse children are more likely to reject them [1].

The support of mothers, fathers, or guardians towards their children’s gender identity is extremely significant for lessening suicidal tendencies among transgender youth. This support also lessens physical abuse at school [2].

For transgender children and adolescents, the lack of support from their guardians, particularly their fathers, was found to be associated with an increase in domestic physical abuse [2,3].

Guardians play an essential role in facing transphobia, protecting and strengthening trans children in the expression of their identity. These guardians need to be educated and supported in the path towards recognizing and managing their own feelings [2] as well as in dealing with the challenges that arise within their networks, especially in schools and community spaces, and from healthcare providers [4,5,6,7].

These networks can contribute to strengthening emotional, face-to-face, instrumental, informational, and self-support support, in order to provide changes from a situation of dependence to one of the autonomies of the target audience [8,9].

Although guardian support is sought, it is scarcely offered. However, these findings come from studies around the children, which justifies the importance of targeting parents and guardians directly to know their experiences and perspectives.

Given the above, the present study aimed to analyze scientific evidence on the dynamics of secondary social networks in supporting mothers, fathers, or guardians of transgender children and adolescents. As recommended by Page et al. [10], the research was registered on the platform “PROSPERO: A registry for systematic review protocols” (registration: CRD42022301747). From a preliminary search in MEDLINE, Cochrane Database of Systematic Reviews, JBI Evidence Synthesis, and PROSPERO, no protocol or review records on the topic were identified, which highlights the innovative potential of this study in relation to the field of scientific evidence available.

## 2. Materials and Methods

The present study is a systematic review of qualitative studies that is a continuation of a review entitled “Dynamics of primary social networks to support mothers, fathers, or guardians of transgender children and adolescents: a systematic review”, previously developed by the authors of this paper and published in this journal (*International Journal of Environmental Research and Public Health*) [5].

This review was guided by the Joanna Briggs Institute Manual for Evidence Synthesis—Systematic reviews of qualitative evidence [11] and Preferred Reporting Items for Systematic Review and Meta-Analysis (PRISMA) recommendations [10]. Qualitative studies examine a given phenomenon in depth in order to analyze nuances that are not achieved in other types of studies, which is essential for the understanding of social networks and their aspects in the health field [12].

The development of this review took place in six stages: theme and guiding question identification; establishment of inclusion and exclusion criteria; definition of information to be extracted from selected studies; assessment of studies included in the systematic review; interpretation of results and synthesis of knowledge [13].

Following the steps, the study’s guiding question was established: “What is the scientific evidence on the dynamics of secondary social networks in supporting mothers, fathers, or guardians of transgender children and adolescents?” This question was defined through the PICo strategy: P (population: mothers, fathers, or guardians of transgender children and adolescents); I (phenomenon of interest: social support); Co (context: social network). In accordance with the social network theoretical framework [8], the guiding question listed studies that responded to the investigation of the dynamics of secondary networks with the description of the relational phenomena present in the networks: alliances; conflicts; discontinuity; ruptures; wear and tear; and transgressions. Thus, it was possible to identify the relationships and functions that compose it.

Original qualitative articles, in all languages, that deal with the social network dynamics through experiences verbalized by the mothers, fathers, or guardians of transgender children and/or adolescents were included. Children and adolescents were defined as people up to 19 years of age, according to the World Health Organization (WHO) classification [14]. Excluded literature comprised the following: duplicate publications; gray literature (abstracts published in proceedings, newspaper news, dissertations, theses, book chapters, letters to the editor, and pre-print publications); other types of study (non-qualitative); studies whose results from the population of interest were not presented separately from other populations (in addition to parents of children and adolescents, including parents of young adults; in addition to parents of transgender people, including parents of LGBTQIA+ people (lesbian, gay, bisexual, transgender/transvestite, queer, intersex, and other identities/orientations); and parents, interviewing children); and papers which did not present results on the social network dynamics.

The article search was carried out in December 2021 through the Central Library system of the University of São Paulo and CAPES journal, which provided access to the following databases: EMBASE, Scopus, MEDLINE, Cumulative Index to Nursing and Allied Health Literature (CINAHL), PsycInfo, Latin American and Caribbean Literature in Health Sciences (LILACS), and Web of Science. For the searches, neither year of publication nor language limits were used.

The search strategy was composed of controlled and free vocabularies combined by Boolean OR operators to distinguish them and Boolean AND operators to associate them, in order to integrate and direct the maximum number of studies on the subject. The same search strategy was readjusted for each database according to their specificities—see published protocol [15].

After surveying the studies in the databases, they were transferred to the Rayyan QCRI online platform [16] for the due exclusion of duplicate studies and subsequent reading of titles and abstracts by two independent researchers, and a third evaluator, for decision in cases of disagreement or doubt between the first two. Therefore, the studies selected in this first stage were submitted to full reading, which allowed analyzing their relevance in relation to their inclusion in the review. The study selection process is presented in the flowchart according to the Preferred Reporting Items for Systematic Review and Meta-Analyses (PRISMA 2020) recommendations [10].

Data extraction was performed using a form adapted from Lockwood et al. [11], composed of author, year and journal of publication, study description (method, phenomenon of interest (objective), study site, participants, data analysis, results, conclusions, and comments). An assessment of the methodological quality of studies included in this review was performed using a checklist for assessing qualitative research proposed by The Joanna Briggs Institute [11].

The results were submitted to a qualitative synthesis in the light of the social network framework proposed by Sanicola [8]. The listed category and the subcategories were validated through discussions with research groups with expertise in the framework, method, and theme in order to ensure the necessary rigor for the discussion of results.

## 3. Results

The study selection step is detailed in the flowchart in Figure 1, which indicates the survey of 9447 publications in the databases and the final selection of 28 qualitative articles to compose the sample of this study.

The articles were published in 2009 (3.5%) [17], 2011 (3.5%) [18], 2013 (3.5%) [19], 2014 (7.1%) [20,21], 2015 (3.5%) [22], 2016 (7.1%) [23,24], 2018 (7.1%) [25,26], 2019 (17.8%) [27,28,29,30,31], 2020 (17.8%) [32,33,34,35,36], and 2021 (28.5%) [37,38,39,40,41,42,43,44]. All articles (100%) were published in English, of which 17 (60.7%) [17,18,21,22,23,24,25,26,27,28,29,32,33,34,35,37,40,44] originated in America, 3 (32.1%) [20,30,31,35,36,37,38,39,41,42] originated in Europe, and 2 (7.1%) [19,43] originated in Oceania. The information synthesis and the main results of the articles included in the final sample of this review are presented in Table 1.

The methodological quality of the articles included in the final sample of this review was assessed and they all showed congruence in the following aspects: philosophical perspective and research methodology; research methodology and research question or objectives; research methodology and data collection methods; participants and their voices adequately represented; compliance with ethical issues and conclusions of data analysis or interpretation [20,21,22,23,24,25,26,27,28,29,30,31,32,33,34,35,36,37,38,39,40].

Regarding the studies’ limitations, it was identified that congruence between research methodology and data representation and analysis was not clear [17] nor presented [27] in two of the studies. There was no clear congruence between research methodology and interpretation of results in two other studies [34,44]. In 11 studies, there was no clarity in the statement that locates the researcher culturally or theoretically [17,18,19,23,25,27,29,34,36,43,44]. In one study, the researcher’s influence on the research and vice versa was not addressed [30]. These results are summarized in Table 2.

## 4. Discussion

The struggle for the transgender cause emerged in studies carried out with mothers, fathers, and guardians of transgender children and adolescents. This took place through coping with barriers to recognizing gender identity and extended to the search for emotional, face-to-face, instrumental, informational, and self-support support to guarantee rights in the context of secondary social networks that participate directly or indirectly in the lives of their children.

In the studies included in this review, the secondary networks determining support were healthcare services [17,18,19,20,21,22,23,24,25,26,27,29,31,32,33,34,35,36,39,40,41,42,43,44], schools [17,18,19,20,21,22,23,24,25,26,27,28,31,35,38,41,42], peer groups [31,36,37,41], parents’ work environment [25,32], and associations (third-sector networks) [35]. Other studies also mentioned the importance of support from religious institutions [19,35], and the non-recognition of religious leaders, whose micro and macro social aggressions were related to the lack of protection and risk of suicide of their children [25,41].

The close contact with children contributed to recognizing gender identity disclosed in childhood and adolescence [32], in which parents were informed directly by the children who insisted that they were of another gender [18], and despite the overload and misinformation, parents reported that they sought support [32]. In this thematic category, secondary social networks are presented in subtopics to facilitate a better understanding.

### 4.1. Healthcare Services

In the health field, it is emphatic in some studies that healthcare services are unprepared for the care of transgender children and adolescents at all levels of care [18,20,27,39], with the non-use of the social name and the lack of information for parents on issues such as hormone therapy [27]. Parents demand professional awareness, so that healthcare professionals are trained, able to answer questions, and can provide adequate assistance [18].

Some studies showed that participants reported recognizing their transgender children and committing to their wellbeing in order to protect them from bullying, depression, anxiety, self-harm, and suicide [22,34,41]. Parents found it difficult to identify local services that could take care of the demands their children had, sometimes associated with suffering in relation to the gender assigned at birth. In this sense, parents felt confused and sought professional help to offer a “diagnostic path”, understanding of the transgender origin and nature, and access to “treatment” options [39], in addition to vaccination clinics where gender identity was recognized [22].

Parents who participated in a study developed in Canada reported barriers in the clinics’ bureaucratic processes, such as waiting times for the start of hormone therapy and the lack of therapists for their children, which generated greater tension in parents [31]. The barriers to care culminate in the anguish of parents due to the potential emotional suffering of their children due to the delay in hormone therapy. Parents perceived themselves to have a role in promoting their children’s wellbeing through direct communication, emotional support, and bonding with healthcare professionals [32].

In another study, it was mentioned that, although medicalization has facilitated access to services, for many, it remained given the implicit suggestion of a possible “treatment” or even a cure. Most participants tended to agree that it would be preferable not to medicalize their children. Many parents shared that their children felt depressed or lonely [22].

Transgender identity is not a disease [23,26,42]; however, biopsychosocial factors can trigger intense suffering and disorders of psychic origin, such as depression, anxiety, and risk of suicide, especially because society does not understand it and because of the frequency of instances of transphobia [22,34,41].

The demands of transgender children and adolescents need to be incorporated into comprehensive healthcare at all levels of care, in addition to offering specialized services for referrals when necessary. Transgender people have both general and specific health demands. Thus, professionals need to be able to welcome a person holistically in the face of their needs, and provide calm and respectful assistance, since being transgender is neither a pathology nor an isolated condition, but a way of being and existing that needs to be recognized and naturalized.

A study carried out in the United States of America with parents of children with gender variance referred to the recommendations from healthcare professionals to discourage parents from allowing their children to express their gender. However, parents perceived that this attitude generated conflict and tension for their children [17].

In this regard, another study carried out in the United Kingdom revealed the importance of valuing the experiences and demands of transgender people and their families, and promoting their participation in health planning. However, the professional imposition of inappropriate behavior was more frequent in parents’ experience, in addition to counseling that reduced the reports from their children, leading to a situation in which parents “did not take it seriously” [27].

Support from healthcare professionals, both emotional and informative, and from other parents (in person and online) was mentioned in a study carried out in Canada. These behaviors contributed to the parents’ understanding of the situation and their role. However, other parents reported feeling excluded from decision-making processes and some described communication problems with healthcare professionals and denial of timely access to necessary care [32].

Gender variation is still perceived by some professionals as a pathology [23]. A study carried out in Italy revealed that parents educated healthcare professionals on the transgender issue, which generated negative, conflicting interactions (fights) between parents and professionals. Specialized services were essentially located in hospitals, where the ambience was reminiscent of illness [42].

In a study carried out in a children’s hospital in the USA, parents/caregivers were hesitant about hormone therapy for their children for fear of possible long-term risks, but they agreed to start hormone therapy after seeking information from professionals on the internet and from personal contacts. The guidance of healthcare professionals contributed to reducing parents’ anxieties [29].

Studies have presented informational support from healthcare professionals for recognizing gender identity by parents [23,34,41]. In this process, therapists are a fundamental member of the support network for counseling, as long as they understand the needs of children and parents to develop their role [40].

Parents need professional support to understand that they are not to blame, that it is okay to stand up for their children’s gender identity, and that there are other families dealing with similar issues. As children enter puberty, parents expressed the need for a “solution” to promote their children’s wellbeing and expressed the need for speaking publicly about the needs of children with gender variance [19].

### 4.2. Peer Groups

Questioning the concepts of “being a man” or “being a woman”—with reference to a given social, cultural, and historical structure—demands an approximation with other experiences, criticalities, and exchanges of knowledge. A study carried out in Italy revealed that, for parents, being in the LGBT+ community was a means of obtaining information, but it did not contribute to the early identification of how their child was experiencing their gender identity [40]. In this regard, the knowledge produced in the LGBT+ community needs greater reach and a social structure that provides a legitimate recognition of gender variations as ways of being and existing.

Groups of peers and/or the friendships between them proved to be important for parents as these allow the sharing of experiences, help in the process of inner change among parents, and the understanding of children as complex human beings—not restricted to gender [19,20,31,35,41,44]. A study carried out in a Pediatric Hospital in Italy with 12 months of group meetings contributed to a decrease in isolation and an increase in self-confidence and hope to support their children. Parents changed their minds about showing their children courage, love, and understanding, and emphasized the importance of welcoming and supporting them. Their participation in the parent group showed positive results by forming a support network and sharing experiences [36].

Community support and strong peer group identification can mitigate stigma [19,44]; contact with other parents of children with gender variance and access to support groups, both face-to-face and online, have been identified as necessary for helping parents (S24, S25) [19,31]. A study carried out in Canada revealed that contact with other parents who went through the same process and meeting in groups were essential factors for recognizing their children’s gender identity. Other participants reported that support from healthcare professionals was more important and that they sought out therapists to help deal with their own emotions [31].

The peer group can be developed and coordinated by mothers, fathers, or guardians of transgender children and adolescents, with activities restricted to members or open. In such peer groups, the exchange of experience, open dialogues or dialogues directed to preestablished themes can be offered with the participation of guests, as well as the support of interdisciplinary professionals able, for instance, to accommodate rights and health demands. These groups can also be organized by a reference professional and have the participation of health teams or a specific specialty, such as therapeutic groups.

A study carried out in France presented the experience of La Salpetrière Hospital in Paris, which carries out a “multi-family” therapeutic group with transgender parents and adolescents of two–twenty families. In these groups, adolescents show humor and laughter, even when painful situations are discussed. The parent group is less harmonious—while some trivialize their family situation, others evoke their anxieties and feel supported and safe, speaking in the group, and sometimes they are emotional. Some left the group after a few sessions, while others remained [43].

The concern of parents with their children’s bodies is recurrent, doubts include the following: What are the postoperative side effects? How do we assess the risk of failure? What if they want to reverse the operation? Do we know enough about the undesirable effects of puberty blockers or hormones on bones? The theme of grief due to children being idealized by parents also emerged. Verbal exchanges between parents mediated by therapists provided satisfactory results in coping with psychological distress or difficulties in family relationships [43]. The respectful bond between tutors (parents and caregivers) provides compliance and strong support to healthcare, in addition to reducing stigma and increasing the comfort of tutors [44].

The group of peers and/or friendship between them proved to be important for parents as it makes it possible to share experiences, help in the process of inner change for parents and the change in their understanding of their children as complex human beings—not restricted to gender [20,36,41].

In a study developed in Canada, parents reported concern about their children’s safety due to the context of transphobia, and reported feeling isolated from other parents because they did not have an empathic feeling for what they were going through. Parents stated that the group was important for their understanding of their children’s transition, supporting them and themselves in this context [37].

A study carried out in Italy also mentioned the benefits of group meetings, which were concerned with the future of their children. Parents changed their minds on the issue of showing their children courage, love, and understanding, and emphasized the importance of welcoming and supporting them. The 12 months of group meetings helped parents to reduce isolation in relation to the issue of their children’s gender identity and to better understand and deal with these issues. Parents’ participation in the parents’ group showed positive results through the formation of a support network and the sharing of experiences relevant to the parents’ experience. They felt less lonely, more self-confident, and positive and hopeful about supporting their children [36].

Another study carried out in Italy described the needs of parents and referred to the importance of a peer group to receive care and later welcome other parents. Group participation changed parents’ perspectives on gender issues, as they reflected on the discrimination LGBTQI+ people are experiencing after witnessing their children’s experiences [42].

### 4.3. Children’s Schools

The school environment is configured as a secondary network that participates in the formation process of individuals, ranging from the learning of basic disciplinary contents to forms of interaction and establishment of norms for social life. Thus, school environments have an important role in the conduction of principles that guide and shape interpersonal relationships. However, in the context of gender variation, traditional school conduct, in which activities are divided in a binary way between female and male, sometimes limits and excludes transgender children from their full development.

Studies carried out in Canada sought to understand the needs of parents of transgender children and adolescents in the development of their children, and the unpreparedness of schools in supporting and protecting these children was reported [22,26]. The school has a critical role [35]; however, there is no recognition [20], no teachings about gender diversity, and no bathrooms for transgender children, with activities and spaces segregated by gender being predominant [24,27], leading families to relocate [20].

Some parents in Italy reported that in kindergarten and elementary school the context was more welcoming; however, binary curriculums for activities, principals’ resistance to adjustments for fear of the reaction of other parents, negative experiences in high school, and lack of support from teachers were reported. Despite parents requesting the use of their children’s social name at school, these requests were not met and parents had to appeal to the law, presenting constitutional articles at school, with the support of peer groups for referrals [42].

In a study carried out in the USA, it was mentioned that parents struggle to create an environment that is conducive to education when guiding teachers and other educators on the needs of their children with gender variation. Parents report that they were less accepted when their children were nonbinary, and they reported an avoidance of unsafe spaces where others could judge their parenting in relation to their children. Furthermore, they reported fear of stigma from other parents who reported that gender variation could influence their own children, and mentioned uncertainty about the future as a source of stress—fearing hormonal transitions and the world’s intolerance towards their children [23].

Participants in a study carried out in the United Kingdom reported parents’ search for information, articulation of knowledge, and assessment of their children’s needs at school, with some being welcoming and others resistant to adapting to the demands. The needs listed included recognition of gender expression, choice of uniform, pronouns, and use of changing rooms and bathrooms. School nurses and psychologists showed broad support [28].

In Canada, parents reported being very concerned about de-pathologizing their children’s transgender identities and expressed extreme concern for their safety due to the constant risk of transphobia at school from peers and teachers, in addition to the binary curriculum structure [26]. Integration between parents and teachers is essential for school reorganization that encompasses transgender students’ gender diversity and other students’ education on the subject [28].

The changes in the school cycle (from preschool to elementary/high school) were considered moments of tiredness by parents of one of the studies, as new schools could be unwelcoming; in addition, bullying was frequent and generated fear that suicidal feelings in children would increase. Parents with transgender children were also asked by parents of other children about supporting the transition and about feeling their children were being subjected to gender diversity. In this context, parents put themselves in the role of tending to their children’s feelings and encouraging them to go to school [28]. In addition to bullying, transgender children have also received death threats. Parents encouraged their children to respond to the offensive jokes of their schoolmates, encouraging them to take a stand in these situations [17].

A study carried out in Australia identified verbal and physical violence at school during the reports of mothers of transgender children. These children suffered from isolation and superficial schooling practices and pedagogies, with supposed support. The school proposed that children use the bathroom for the disabled, and the offer of protection from harassment took place through the removal of transgender children from their classroom and activities—behaviors that contributed to greater social isolation. Support for these children is seen as a burden by schools and not as an opportunity for criticism and growth [38]. The school was referred to as an ineffective support network, with an impact on social, emotional, physical, and academic wellbeing [20,38].

### 4.4. Other Social Networks and Sociopolitical and Cultural Contexts

Discrimination against transgender children and adolescents was mentioned in a study carried out in the USA as coming from other parents or unknown community members. In the same study, it was also found that society was uninformed about the needs and challenges faced by parents, including discrimination in travel contexts related to ticket issuance, in which identity documentation should be presented; in addition, cases of misgendering were reported, in which people are assigned a gender that does not correspond to their identity—either intentionally or apparently inadvertently [30].

Another important social network listed by the participants of a study carried out in the USA is the institution/workplace of the parents of transgender children and adolescents. The study reported the importance of a supervisor’s understanding and authorization for parents to accompany their children to medical appointments [35]. Another study pointed to associations, third-sector networks, as facilitators in the positive evolution of the social transition process [33].

In the field of politics, government, and legislative support, parents expressed the relentless day-to-day problems of realizing that their child—in addition to being marginalized—is also denied their rights in school, policies, government legislation, and health resources. Additionally, happiness, success, fulfillment, peace, security, and supportive personal relationships were variously mentioned by parents as a “right” of their children. Parents’ needs vary depending on their level of knowledge, their child’s age, and how recently they became aware of their child’s gender variance [18].

Other relevant factors are community attitudes and the current political climate, finances related to children’s gender transition, participants’ hopes and concerns about acceptance by others, safety, and community attitudes. In one of the studies, parents expressed difficulty concerning the level of support and the lack of understanding from hairdressers, nurses, teachers, and other parents/caregivers. There were difficulties with their children’s legal gender transition, both personally and in relation to the sociopolitical climate in the USA [41].

Parents report that supportive cultures and societies that do not impose or reinforce stereotypes are critical for ensuring consistent support for their children. Such a culture/society might involve contact with transgender people (visibility and positive portrayals of transgender individuals and communities) and financial, legal, and government support (paying counseling and other professional expenses; politicians and leaders aware of the problems faced by parents of children with gender variation) [19].

It is urgent to understand the process of gender development in transgender children and youth. This will help improve clinical, social, medical, and educational interventions to better serve young people and their families and overcome the different levels of oppression they face. Support from local and wider community networks includes assistance and acceptance from local faith groups, clubs, and schools, and more tolerance for gender variation in society at large [19].

It is important to mention that support for those responsible for trans children and adolescents does not replace comprehensive care and listening to said group, but must occur in a complementary way. The broader scientific literature also highlights the risks of transphobic violence perpetrated mainly by those responsible, in addition to teachers and healthcare providers [2]. These studies are generally carried out with children and adolescents for an overall understanding of the problem.

In general, the methodological quality of the studies included in the final sample of this review were satisfactory; however, it is recommended that the identified limitations be considered with regard to obtaining greater clarity and advocating for the inclusion of relevant information. Thus, studies need to pay attention to clarity between method and data analysis, interpretation of results, statements that locate the researcher culturally or theoretically, and consider the researcher’s influence on the research, so that qualitative nuances are better understood and contextualized.

The present study had limitations, such as the non-inclusion of gray literature and the lack of a manual study search. On the other hand, this study considered that original articles published in journals indexed in the elected databases and identified from the formulated search strategy were sufficient for envisioning the proposed investigation and prioritizing the studies’ methodological quality. Moreover, the decision to include only original articles allowed the inclusion of research results that have greater theoretical and methodological rigor—ensured by expert reviewers.

## 5. Conclusions

The studies included in this review presented the challenges faced by mothers, fathers, or guardians of transgender children and adolescents in achieving rights-based support, especially in social, healthcare, and education services, in addition to informal exchanges and links.

Secondary social networks, especially healthcare services and children’s schools, proved to be potential networks to strengthen bonds for emotional and informational support; however, the literature shows a lack of preparation among professionals and institutional policies to welcome transgender children and adolescents and their families.

On the other hand, peer groups stood out as an important support network for mothers, fathers, and guardians, whereby members can meet and share their anxieties, exchange experiences, and acquire knowledge.

The information synthesized in this review can contribute to the development of resources and measures to strengthen ties that consider transgender demands in health institutions, schools, and the community, in addition to influencing the development of policies and proposing future research on health promotion.

## Figures and Tables

**Figure 1 ijerph-19-08652-f001:**
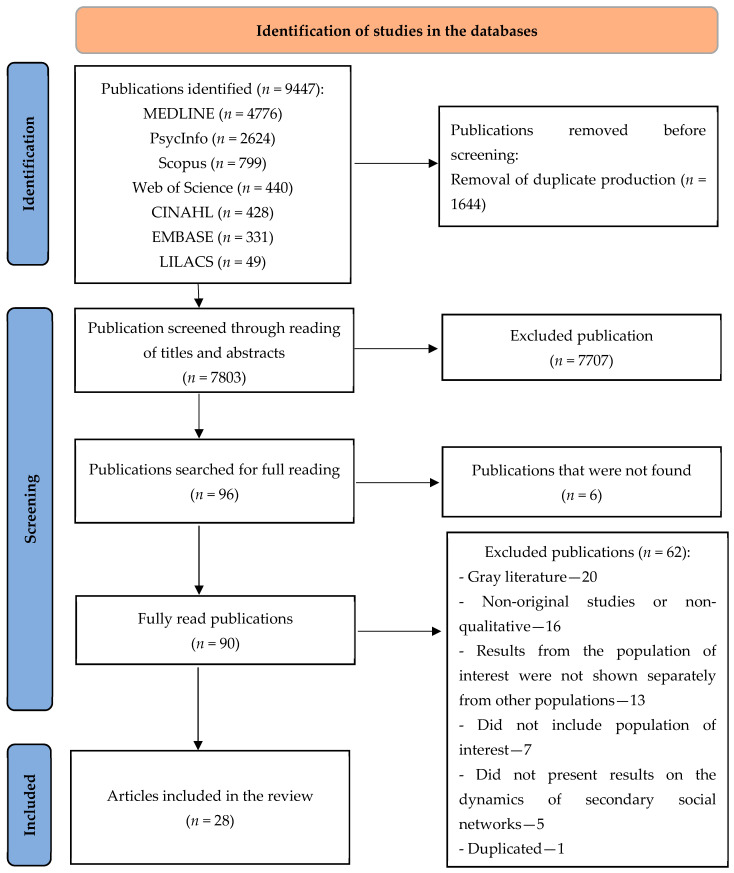
Flowchart of studies selected for systematic review on the dynamics of secondary social networks to support mothers, fathers, or guardians of transgender children and adolescents. Ribeirão Preto, SP, Brazil, 2022. Source: Adapted from Page et al. [10].

**Table 1 ijerph-19-08652-t001:** Information and synthesis of the main results of studies selected for systematic review on the dynamics of secondary social networks to support mothers, fathers, or guardians of transgender children and adolescents. Ribeirao Preto, SP, Brazil, 2022.

ID	Author/Journal/Year/Country	Objective	Study; Sites; Participants	Data Collection and Analysis	Main Results
S1 [17]	Hill; Menvielle/*Journal of LGBT Youth*/2009/USA	To report problems faced by parents of children with gender variation behaviors in childhood and/or gender variation identity and compile their knowledge.	USA and Canada; 43 parents (heterosexual and lesbian couples) of 31 transgender children.	Semi-structured open-ended interviews through the phone; analysis—not applicable.	Physicians and psychologists in some cases encouraged parents to discourage gender variance behavior in their children and said it would be just a phase. Parents started a movement to police their children’s behavior, making them behave more according to their birth gender. They noticed that this generated conflict for children and they were even more tense by this policing. It was reported that, at school, children experienced bullying and even death threats; they stated that the involvement of educators in the process of including children with gender variation and the process of creating an anti-bullying school would be important factors. Parents encouraged their children to take a stand against the offensive play of their classmates.
S2 [18]	Riley et al./*International Journal of Sexual Health*/2011/USA	Provide a foundation to support all children with gender variation and their parents by identifying their needs.	USA, Australia, Canada, and the United Kingdom; 27 mothers, 3 fathers, and 1 guardian of transgender children; snowball sampling was used.	Interviews via internet with closed-ended and open-ended questions; content analysis, with the help of Weft qualitative data analysis (QDA), with a continuous reflective–interpretative process to generate the themes.	Emotional support was mentioned by parents in the form of professional support for counseling for themselves, their families, and their children, and the importance of support groups and other transgender individuals also emerged. Parents reported the need for more awareness and preparation of healthcare professionals, expressing frustration with the scarcity of available medical support and the difficulty of access to information and health. Parents expressed the relentless daily problems of realizing that their child, in addition to being marginalized, is also denied their rights in school, policies, government legislation, and health resources.
S3 [19]	Riley et al./*Sex Education*/2013/Australia	Investigate and understand the experiences of people who have the experience and knowledge needed to determine the needs of children with gender variation and their parents.	Sydney (Australia); parents with children aged 12 years and below, transgender adults, and clinical professionals with experience working with the transgender community; snowball sampling was used.	Online interview with closed-ended and open-ended questions; Zoomerang method was used; grounded theory and content/thematic analysis were used, which involved an interpretative process reflection; coding method was used, following Buckingham and Saunders.	The needs of parents of children with gender variation include: information; education (identified as a necessity for counselors and medical professionals, for school staff, for parents, and for community programs); professional advice and support; peer and community support; contact with transgender people (visibility and positive portraits of transgender individuals and communities); and financial, legal, and government support.
S4 [20]	Kuvalanka; Weinera; Mahana/*Journal of GLBT Family Studies*/2014/USA	To understand how parents of transgender children begin to identify diverse gender expressions in their children, how they feel about their children’s expression, and to understand how their social contexts have impacted family experiences.	Oxford, Ohio, (USA); 5 mothers of transsexual children aged between 8 and 11 years recruited through social networks and contacts of professionals from the study advisory board.	Interviews through the telephone; open coding (inductive thematic analysis); three main thematic fields emerged.	Social acceptance and appropriate medical intervention for transition were important for children’s wellbeing. Mothers reported that the real transition was theirs and not of their children, because they only materialized what they always were internally. Some healthcare professionals advised mothers that gender-diverse behavior was a phase and even blamed mothers who left their children free with what they wanted to play/dress; others, despite not being experts, sought information to better attend to the cases. Overall, the school showed little acceptance, isolated children socially, and bullying was practiced by other children. Parents were isolated by their communities who did not understand/accept the transition of their children, causing the family to change places. Caregivers reported finding relief in the support of people who went through the same process.
S5 [21]	Platero/*Journal of GLBT Family Studies*/2014/Spain	To raise awareness about the lack of knowledge and trigger an informed discussion to analyze how families talk, behave, and feel about their young children and about themselves.	Madrid (Spain); 12 parents (9 mothers and 3 fathers) of transgender children and adolescents.	Face-to-face and telephone interviews; thematic analysis, performed by identification of codes and themes.	Parents indicated that most of the information they received was from the internet, hospitals, psychologists, and medical staff. Only a few had information from homosexual and transgender services (public services available) or, more rarely, through social services, LGBT parenting organizations, or schools. Parents often reported that they had to “educate” the professionals around them, such as teachers, general practitioners, and social workers. Most searched for a long time and without success for professionals who could provide adequate information, sometimes traveling to different regions of Spain. Developing a good relationship between parents and professionals made it possible to include perspectives on their children in the interviews.
S6 [22]	Sansfaçon; Robichaud; Dumais-Michaud/*Journal of LGBT Youth*/2015/Canada	Understand the problems and challenges experienced by parents of children with gender variation in the process of supporting gender identity and expression of their children as they grow up.	Montreal, Canada; 14 parents of children with gender variance.	Participatory action research using the social action methodology (SAM) principles and processes and focus groups; grounded theory; data were analyzed as they were collected and involved open, axial, and selective coding.	Although medicalization facilitated access to services, for many, it remained given the implicit suggestion of a possible “treatment” or even cure. Another challenge was working with schools to support and protect their children, as schools were poorly prepared or insufficiently qualified and found it difficult to identify or find specialized services to work with their children as a health center or vaccination clinics for everyday issues.
S7 [23]	Gray et al./*Family Process Institute*/2016/USA	Describe the experience of parenting of a child with gender variation as well as the mutual influence between the child, the family, and the environment.	Boston, Massachusetts (USA); 11 caregivers, 8 mothers, and 3 fathers.	Interviews with semi-structured questionnaire; qualitative approach based on grounded theory, creating units of meaning, and comparing them to identify categories based on similarity.	Parents described positive experiences of support from the LGBTQ community that contributed to trust. The lack of support of healthcare professionals who dealt with the issue as a pathology to be “cured” was also mentioned. In this study, it was mentioned that parents of children with gender variation struggle to create a “normal” childhood and an environment conducive to education, guiding teachers and other professional educators about the needs of their children with gender variation. Advocacy included educating teachers and school administrators.
S8 [24]	Pyne/*Journal of Progressive Human Services*/2016/Canada	Focus a lens on parents of transgender children who affirm their children’s sense of gender, explore how these parents know their children’s gender identities and develop a theory to better understand the knowledge underlying the decision to assert children’s self-identities.	Canada; 15 parents of children and adolescents up to 12 years of age, nonconforming gender.	Semi-structured interviews; grounded theory, performed in open, axial, and focused or selective coding to incorporate the categories.	Parents’ conflicts included concern about structural contexts such as the predominance of activities and spaces segregated by gender in schools, such as bathrooms. Some parents found mental healthcare professionals to be unprepared, who referred them to another place or misunderstood their role, in addition to language limitations.
S9 [25]	Alegría/*International Journal of Transgenderism*/2018/USA	Understand the experience of parents and caregivers of transgender children/adolescents and their relationships with close family members.	Six American states (four on the west coast and two in the east); 14 fathers/mothers/guardians.	Semi-structured interviews; inductive analysis using a comparative method; emerging themes were identified.	Establishing support networks and participating in support groups provided resources for education and resilience. The support groups mentioned were the following: local and regional; remote; school and health advocacy; unpreparedness of healthcare professionals in the approach and conduct of care; family willingness to leave church and change schools.
S10 [26]	Newhook et al./*Canadian Journal of Community Mental Health*/2018/Canada	Examine the needs and concerns of transgender youth and their families across the island of Newfoundland.	Newfoundland (Canada); 21 parents/guardians and 24 transgender adolescents. Most of them were not from the same family.	Individual interviews through electronic questionnaires, via Google Forms; thematic analysis of participants’ verbalizations.	Participating parents felt comfortable talking to the family physician about their children’s gender identity; however, young people felt uncomfortable with these discussions. Parents and young people expressed concern about the lack of knowledge about health for transgender young people among general practitioners. Parents’ and adolescents’ and children’s concern include: (1) waiting time for care; (2) their children’s mental health; (3) lack of information or guidance; (4) safety and transphobia at school; and (5) de-pathologizing children’s identities.
S11 [27]	Carlile/*International Journal of Transgenderism*/2019/The United Kingdom	Investigate the experiences of transgender children and adolescents and their families in their interactions with primary and secondary healthcare providers in England.	England, the United Kingdom; 65 transgender and nonbinary parents and children between 12 and 18 years of age and the remaining adults.	Participant–researcher model, here called “Illuminate”; analysis with identification of themes and sub-themes outlined and highlighted in a document.	Healthcare professionals were considered unprepared and without adequate knowledge. Misgendering and deadnaming experiences have been attributed to a lack of education about gender identity issues. Waiting lists and long journeys to healthcare were mentioned. Negligence with the service user’s expertise was evidenced, especially in relation to what can help at school and the lack of patient and family participation in clinical and therapeutic planning. These aspects were expanded in situations where transgender children were autistic. Parents felt harmed and without support from family healthcare professionals and children and adolescents healthcare professionals. This generated mental and emotional suffering for the family, and the parents mentioned “we are also patients”. Some professionals advised parents to “not take it seriously”, and to ignore or punish self-injurious behaviors related to body dysphoria.
S12 [28]	Davy; Córdoba/*GLBT Family Studies*/2019/The United Kingdom	Understand the experiences of parents who support their children within school cultures.	United Kingdom; 23 parents of transgender and gender-diverse children from different schools across the UK.	Individual interviews via Skype or in person (at the participant’s residence); thematic analysis with a deductive approach; data coding according to thematic axes was analyzed separately by the authors and together, giving rise to new codes and themes, with the help of NVivo 11 software.	Gender processes within the school system were seen as a borderline situation by parents, where teachers may be reproducing and reinforcing gender processes, but these were seen as mutable through dialogic strategies through the generation of what Paulo Freire called critical consciousness. Parents sought as much information as possible (interacting with peer groups and support organizations) to prepare the school. Each parent took on an oppressive (school) culture that their children could pass through, and the limit situations were difficulties in the use of name/pronoun, antagonistic employees and parents, spaces segregated by gender, and bullying. School support required interaction and dialogue with parents through knowledge from their experiences and the challenge of adapting school cultures.
S13 [29]	Daley et al./*Journal of Adolescent Health*/2019/USA	Understand the decision making of adolescents and parents about the process of gender-affirming hormone therapy.	Cincinnati, Ohio (USA); 17 adolescents aged between 14 and 20 years who started hormone therapy and 13 parents of these adolescents.	Semi-structured interviews; thematic analysis; data were coded in pairs; Nvivo 11 was used for coding and analysis.	Parents/caregivers were hesitant about the hormonization their children wanted to go through. They agreed to start hormone therapy only after seeking information from professionals on the internet and with personal contacts. As parents received informational support, they opened up more to gender-affirmation hormone therapy (GAHT) as an option. They were informed on the internet and on websites that offered medical services. They report the grief they experienced for the loss of their idealized child. They reported that healthcare professionals helped to reduce parents’ anxieties about GAHT.
S14 [30]	Hidalgo; Chen/*Journal of Family*/2019/USA	Explore how, if at all, parents of prepubescent transgender people experience gender minority stress related to their children’s gender identity/expression.	USA; 40 parents, 8 participating alone and 16 participating as a dyad and their children: 24 children and adolescents aged between 4 and 11 years assisted at a gender clinic.	Focus groups, following a scripted protocol; content analysis; a multiphase coding process to establish reliability was employed; Dedoose software was used.	Parents mentioned situations of transphobia in the social context, due to lack of knowledge in society in general and discrimination related to travel, especially at the time of issuing tickets for a transgender child in social transition and when identity documentation must be presented, generating stress for parents.
S15 [31]	Sansfaçon et al./*Journal of Family Issues*/2019/Canada	Explore the journey of parents of transgender children regarding acceptance of their children’s gender identity, including reactions to child transformations, struggles, facilitators of acceptance, and experiences lived in clinical settings.	Montreal, Quebec; Ottawa, Ontario; and Winnipeg, Manitoba. 4 fathers and 32 mothers of 35 children and adolescents from 9 to 17 years old.	Semi-structured interviews; inductive and reflective thematic analysis; transcripts were coded and separated into thematic areas; the data analysis software used was MAXQDA.	Contact with other parents who were going through the same and meeting in groups were essential factors for recognizing their children. Others reported that the support of healthcare professionals was more important, seeking therapists to help deal with their own emotions. Access to specialized healthcare services, in general, was easy, being indicated by people or even schools. They report that the bureaucratic processes of clinics, such as waiting time for the start of hormone therapy and the lack of therapists for children and parents, were barriers that generated tension in parents.
S18 [32]	Clark et al./*Elsevier Journal of Adolescence*/2020/Canada	Explore about the decision of transgender youth and their parents made about the start of hormone therapy.	British Columbia (Canada); 21 transgender youth aged between 14 and 18 years and 15 parents of these young people; snowball sampling was used.	Semi-structured interviews; constructivist grounded theory analysis of interview transcripts and lifeline drawings, performed within groups, then between groups, with the help of NVIVO 11 Pro.	Overall, parents’ experience with the support of healthcare professionals was mixed. Seeking support from healthcare professionals and other parents (in person or remotely) helped participating parents understand the situation and their role. Parents identified emotional support, answers to questions, and information about the process of initiating hormone therapy as components of positive interactions with healthcare professionals. Some parents, however, felt excluded from decision-making processes, and others described problems in communication between parents and healthcare professionals, in one case resulting in a young person being denied timely access to needed care. Many parents have faced barriers to accessing hormone therapy, including a lack of qualified professionals. Parents found themselves in need of research on hormonization, and working with schools on a transition support plan.
S19 [33]	Testoni; Pinducciu/*Sciendo*/2020/Italy	Consider how parents of transgender children dealt with their transition and how they live the experience of grief.	Italy, Spain, and the United States; 18 parents (11 cis-female, 6 cis-male, 1 nonbinary). Spain (5), Italy (6), and USA (7).	Online individual interviews, through the SurveyMonkeys platform; thematic analysis, with the aid of Atlas.ti.	The support of associations formed by another country was essential to hear similar stories, share feelings, and reinforce the awareness that their children have always been the same and have a gender identity that needs to be recognized. Most participants were engaged in advocacy activities and the experiences helped organize their own biographical narratives.
S20 [34]	Thornburgh et al./*Pediatrics*/2020/USA	Describe experiences in partnership with parents of young people with gender diversity to better support our patients and families.	USA; parents, gender-diverse young people, and gender-diverse adults who promoted support to parents of transgender novices in the clinic.	Reports about support experience occurred by phone, email, or in person; description of experiences and presentation of some excerpts of participants’ speeches.	The beginning and end of visits were reported in collaboration, in which a behavioral healthcare professional met with parents (who verbalize their fears and anxieties) while physicians speak with patients alone. It was emphasized that one of the most important factors in raising a child with gender diversity is to connect with other parents of young people with gender diversity.
S16 [35]	Bhattacharya et al./*Journal of Family Psychology*/2020/USA	Understand the perspectives of transgender youth and their caregivers, young individual–caregiver relationships, and caregiver–caregiver relationships in the family system.	USA; 20 families (20 transgender youth aged between 7 and 18 years old and 34 caregivers).	Semi-structured interviews; thematic analysis with immersion and crystallization method; Dedoose program was used.	In relation to caregiver relationships, in the families of mothers and fathers, mothers were more proactive than fathers in accessing gender-affirmation services for their children. Contextual factors that influence youth–caregiver relationships were school, community support, religion, and workplace.
S17 [36]	Caldarera et al./*Clinical Child Psychology and Psychiatry*/2020/Italy	Describe what themes and benefits were perceived regarding the participation of parents responsible for children with gender diversity in a psychological support group set up for parents of young people who attend a gender identity development service.	Torino (Italy); 11 parents of children and adolescents aged 8–17 years.	Reports from group meetings; thematic analysis (familiarization, production of codes, generation of themes, being reviewed, refined, named, and a report was produced).	Peer group support to share concerns and give positive tips raises awareness and sense of serenity, gives access to correct information, and helps people escape from isolation by having mutual support. After attending the group for 12 months, there was a greater understanding of children’s gender identity and a change in the approach to dealing with child gender diversity; benefits of attending the group; perception of the importance of having an understanding, empathetic, and supportive attitude; deep understanding of gender diversity and children’s needs; alleviation of shame and lessening of the sense of guilt; perceived support from other parents and physicians, along with increased self-confidence; and a perceived importance of communication with children.
S21 [37]	Dangaltcheva/*Frontiers in Psychology*/2021/Canada	Describe the adaptation of the Connect Program to meet the needs of parents of transgender and gender-nonconforming youth, and measure the program effectiveness.	British Columbia, Canada; 20 parents (14 mothers and 6 fathers) of 16 young people with nonconforming gender aged between 12 and 18 years.	Group dynamics; the average participation was 9 out of 10 sessions; model analysis, through notes of recorded sessions, reviewed and attributed themes, using NVivo 11.	The Transforming Connections program with group meetings for caregivers of transgender and gender-diverse children (TGD) and adolescents was mentioned by parents as a support strategy for understanding gender identity issues. Emerging themes included: coming out, connecting with peers, asserting pronouns/nouns, transition, parental reactions (confusion, isolation, sadness, and acceptance), and safety and mental health concerns. Caregivers reported feeling respected, safe, and welcomed in the program, improved understanding of children as well as themselves.
S22 [38]	Ferfolja; Ullman/*Pedagogy, Culture & Society*/2021/Australia	Understand what was happening to the participants’ children at school and, more importantly, how these parents read and navigated their children’s educational experiences.	Australia; 10 parents with TGD children.	Online forum interviews and face-to-face interviews; data analysis in NVivo, where they were analyzed using a coding framework developed by the two researchers and an assistant.	Mothers reported the verbal and physical violence that their children suffered at school and lack of discussion on diversity by teachers. Many parents of TGD children spoke of how their child was isolated and marked by school practices and pedagogies that aimed (supposedly) to support students. Inadequacy of bathroom use and isolation of TGD children. Support for TGD students was seen as a burden by schools and not as an opportunity for reflection or growth. Support from schools, at the request of parents, was superficial and ineffective.
S23 [39]	Frigerio et al./*Journal of GLBT Family Studies*/2021/Italy	Explore the experiences of parents of transgender adolescents diagnosed with gender dysphoria who, for the first time, attended a clinic for psychological consultation.	Milan (Italy); 15 parents (10 mothers and 5 fathers) of TGD adolescents, mostly (93%) transgender boys, aged between 14 and 19 years.	Individual interviews via Skype or phone; inductive thematic analysis through coding, search for themes and organization of themes.	Parents highlighted a mismatch between the expectations and needs they identify as priorities for themselves and their children and the support that healthcare services offer (difficulty identifying local services that can take care of their children’s suffering and offer a diagnostic path to understanding its nature; parents found it difficult to understand how to access treatment options).
S24 [40]	Iudici; Orczyk/*Sexuality & Culture*/2021/Italy	Explore how parents develop an understanding of their children’s gender identity, how they stand in relation to this and how the topic is treated in relation to family, social, and institutional contexts and children’s health needs.	Italy; 20 parents of minors who did not identify with their assigned gender.	Semi-structured interviews; discourse analysis, performed through coding and themes.	As a parent begins to understand gender variation, the search for information is first given by an information vehicle (the internet). Some parents resorted to psychological professionals to understand how to behave with their children. For a parent, being in the LGBTQ+ community was an informational resource, but it did not help the parent identify early on how their child was experiencing their gender identity. How parents are coping with daily situations related to gender variation: parents shape the situation they are facing as a civil and cultural “battle”, so as to obtain the civil rights they currently lack. Many parents are also part of peer groups, virtual or otherwise, that support each other.
S25 [41]	Katz-Wise et al./*Journal of Family Issues*/2021/USA	Explore attitudes and challenges faced by parents/caregivers of transgender and/or nonbinary youth.	USA; 27 parents/caregivers of transgender and/or nonbinary children, adolescents, and youth.	Interview through online form; thematic analysis, using immersion/crystallization approaches to identify themes; primary coding was completed using Dedoose software platform.	Participants reported receiving or desiring support from Facebook groups or other online resources, their children’s school, therapy, support groups, friends and colleagues, books, or were unsure. External factors comprised community attitudes and current political climate, finances related to children’s gender transition, participants’ hopes and concerns about acceptance by others, safety, and community attitudes. The disclosure of children’s gender identity and lack of support or need from parents or children and the need for support from professionals to obtain more information emerged. They expressed a lack of understanding from hairdressers, nurses, teachers, other parents/caregivers. There were difficulties in legally transitioning their children’s gender both personally and in relation to the sociopolitical climate in the USA. Moreover, several participants mentioned difficulties with their children’s gender transition and feelings of grief or loss.
S26 [42]	Lorusso; Albanesi/*J Community Appl Soc Psychol.*/2021/Italy	Map/describe the needs of parents of transgender and children with gender variance in Italy, their relationship with the health and education systems, and how they deal with the challenges of the context in which they live.	Italy; 13 parents recruited by the snowball technique; four cisgender fathers; nine mothers; among them: three heteroparental and one female homoparental; their children’s ages ranged from 5 to 17 years.	Semi-structured interviews, individual, via Skype; reflective thematic content analysis, in which the first themes were generated, being validated by participants by email for building the final themes.	Parents were critical of Italian society and Catholic tradition as an obstacle to obtaining civil rights and disseminating accurate information. Parents looked for a pediatrician or psychologist. However, parents who had to inform professionals about these issues, which also generated negative and conflicting experiences between parents and professionals and specialized services located essentially in hospitals, as the ambience referred to the idea of illness. School system offers binary curriculum and lack of teacher support. The importance of support groups for parents emerged.
S27 [43]	Rabain/*Transgender Adolescents and Their Parents*/2021/France	For parents: Reflect on family relationships and deal with all types of experiences of discrimination.	Paris, France; parents and adolescents in support groups gathering in two–twenty families.	Group interviews with progressive inclusion; identification of recurring themes arising from group therapeutic approaches for parents and adolescents and for both (multifamily).	In therapeutic groups formed by parents, some trivialized their family situation, others evoked their anxieties and felt supported, safe to speak in the group, and sometimes emotional. The concern for their children’s bodies was recurrent. Co-therapists invited group members to talk about anything that comes to mind, “even if it’s not easy”, as verbal exchanges also provide very solid support.
S28 [44]	Szilagyia; Olezeskib/*Smith College Studies in Social Work*/2021/USA	Discuss unique challenges encountered in working with parents and caregivers of transgender adolescents during virtual visits that have the potential to interfere with the development of a therapeutic alliance and the movement towards greater family acceptance.	Connecticut (USA); parents/caregivers of transgender adolescents participated.	Meetings by videoconference of team members with children and guardian, together then separately; description of two clinical cases for interpretation and discussion.	Respectful engagement with both guardians (parents and caregivers) and transgender youth by medical and mental healthcare professionals in the clinical setting of a gender program can help practitioners establish a strong therapeutic relationship and potentially play a role in diminishing stigma and increasing the comfort of guardians.

USA—United States of America.

**Table 2 ijerph-19-08652-t002:** Methodological quality of articles included in this systematic review on the dynamics of secondary social networks to support mothers, fathers, or guardians of transgender children and adolescents. Ribeirao Preto, SP, Brazil, 2022.

	1. Is There Congruence between the Stated Philosophical Perspective and the Research Methodology?	2. Is There Congruence between the Research Methodology and the Research Question or Objectives?	3. Is There Congruence between the Research Methodology and the Methods Used to Collect the Data?	4. Is There Congruence between the Research Methodology and the REPRESENTATION and Analysis of Data?	5. Is There Congruence between the Research Methodology and the Interpretation of Results?	6. Is There a Statement Locating the Researcher Culturally or Theoretically?	7. Is the Researcher’s Influence on Research and vice Versa Addressed?	8. Are Participants and Their Voices Adequately Represented?	9. Is the Research Ethical according to Current Criteria or, for Recent Studies, Is There Evidence of Ethical Approval by an Appropriate Body?	10. Do the Conclusions Drawn in the Research Report Stem from Data Analysis or Interpretation?
Hill; Menvielle, 2009 [17]	Y	Y	Y	U	Y	U	Y	Y	Y	Y
Riley et al., 2011 [18]	Y	Y	Y	Y	Y	U	Y	Y	Y	Y
Riley et al., 2013 [19]	Y	Y	Y	Y	Y	U	Y	Y	Y	Y
Kuvalanka et al., 2014 [20]	Y	Y	Y	Y	Y	Y	Y	Y	Y	Y
Platero, 2014 [21]	Y	Y	Y	Y	Y	Y	Y	Y	Y	Y
Sansfaçon et al., 2015 [22]	Y	Y	Y	Y	Y	Y	Y	Y	Y	Y
Grey et al., 2016 [23]	Y	Y	Y	Y	Y	U	Y	Y	Y	Y
Pyne, 2016 [24]	Y	Y	Y	Y	Y	Y	Y	Y	Y	Y
Alegría, 2018 [25]	Y	Y	Y	Y	Y	U	Y	Y	Y	Y
Newhook et al., 2018 [26]	Y	Y	Y	Y	Y	Y	Y	Y	Y	Y
Carlile, 2019 [27]	Y	Y	Y	N	Y	U	Y	Y	Y	Y
Davy; Córdoba, 2019 [28]	Y	Y	Y	Y	Y	Y	Y	Y	Y	Y
Daley et al., 2019 [29]	Y	Y	Y	Y	Y	U	Y	Y	Y	Y
Hidalgo; Chen, 2019 [30]	Y	Y	Y	Y	Y	Y	N	Y	Y	Y
Sansfaçon et al., 2019 [31]	Y	Y	Y	Y	Y	Y	Y	Y	Y	Y
Clark et al., 2020 [32]	Y	Y	Y	Y	Y	Y	Y	Y	Y	Y
Testoni; Pinducciu, 2020 [33]	Y	Y	Y	Y	Y	Y	Y	Y	Y	Y
Thornburgh et al., 2020 [34]	Y	Y	Y	Y	U	U	Y	Y	Y	Y
Bhattacharya et al., 2021 [35]	Y	Y	Y	Y	Y	Y	Y	Y	Y	Y
Caldarera et al., 2021 [36]	Y	Y	Y	Y	Y	U	Y	Y	Y	Y
Dangaltcheva et al., 2021 [37]	Y	Y	Y	Y	Y	Y	Y	Y	Y	Y
Ferfolja; Ullman, 2021 [38]	Y	Y	Y	Y	Y	Y	Y	Y	Y	Y
Frigerio et al., 2021 [39]	Y	Y	Y	Y	Y	Y	Y	Y	Y	Y
Iudici; Orczyk, 2021 [40]	Y	Y	Y	Y	Y	Y	Y	Y	Y	Y
Katz-Wise et al., 2021 [41]	Y	Y	Y	Y	Y	Y	Y	Y	Y	Y
Lorusso; Albanesi, 2021 [42]	Y	Y	Y	Y	Y	Y	Y	Y	Y	Y
Rabain, 2021 [43]	Y	Y	Y	Y	Y	U	Y	Y	Y	Y
Szilagyi; Olezeskib, 2021 [44]	Y	Y	Y	Y	U	U	Y	Y	Y	Y

Legend: Y—yes; U—unclear.

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
