# Peer review of "Support for Mothers, Fathers, or Guardians of Transgender Children and Adolescents: A Systematic Review on the Dynamics of Secondary Social Networks"

_ijerph, 2022, doi:10.3390/ijerph19148652_

Round 1

Reviewer 1 Report

Overall this is a strong piece with a few main issues that need to be fixed. There are some wording or numbering muddles, some oversights in the picture given of the established research on parents and guardians from studies of transgender and gender diverse youth and how this study relates to that broader literature, and some extra minor details about the study to be included. A few efforts in these regards to address these issues, will really bring this piece together however, and I see it being a useful piece beyond the parent studies readership and potentially into the much bigger trans youth studies readership if the concerns of trans youth studies are better accounted for and compared here.

Abstract

The abstract is a little clumsy and disconnected from existing literature. For example it opens P.1 Line 1: “Recognizing gender identity in childhood and adolescence by mothers, fathers, or guardians is little explored in scientific literature, nor is emphasis given to the dynamics of secondary social support networks that involves relationship ties with socio-sanitary service institutions in health, education and social fields”. The grammar here is poor as this sentence starts with a continuing verb. Also this sentence relies on assumptions the readers have some pre-existing knowledge of or interest in social support networks (as though this is some widely known and agreed collection of undebated categorisation) and so forth. You need to start an abstract with what is known and you also need to be broader in conceiving who may care for trans youth remembering many are cared for by guardians. For example it is known that “Mothers, fathers, or guardians’ support for disclosures of diverse gender identity has significant relationships with decreased suicidality for transgender children and adolescents”. This is an important and well known association that then makes the lack of exploration of recognition more important. The piece should open with it. Then be a bit clearer on what you mean by social networks (professionals, friends or what groups) – you need to be more direct in defining this, as currently definition occurs too indirectly.

Introduction

The introduction inherits the somewhat clumsy opening phrasing of the abstract and the strong disconnection from existing literature. For example it opens P.1 Line 31: “Recognizing gender identity by mothers, fathers, or guardians of transgender children and adolescents can occur immediately after disclosure of their own children or in  the course of the search for answers to their desires and individual reactions associated or 33 not with the feeling of “loss” of an idealized child. Again, the grammar here is poor as this sentence starts with a continuing verb, and it would be better to open with “Mothers, fathers, or guardians’ recognition of gender identity can occur…”, or to start the sentence with something like ‘When transgender children…’.  

The introduction like the abstract also needs to start with a stronger justification about why “Mothers, fathers, or guardians’ support for gender identity” matters… why it should be studied, beyond the mere claim that it has not been studied; given it has been studied and seminal correlations uncovered that have informed parental education groups like PFLAG for years.  It is therefore instead important to note from the existing literature that transgender and gender diverse youth are significantly more likely than cisgender same-sex attracted youth to have disclosed their identity to most people in their lives including their parents, and that fathers are comparatively the most likely to reject transgender and gender diverse youth [Jones, T. & Hillier, L. (2013) Comparing Trans-Spectrum and Same-sex- Attracted Youth in Australia: Increased Risks, Increased Activisms, Journal of LGBT Youth, 10:4, 287-307, DOI: 10.1080/19361653.2013.825197]. Again, it is most directly notable in the existing literature that “Mothers, fathers, or guardians’ support for gender identity disclosures” has highly significant relationships with decreased suicidality [Hillier et al., 2010, Writing Themselves in 3. Melbourne: ARCSHS, pp. 74-75], for transgender children and adolescents.  Mothers, fathers, or guardians’ support for gender identity disclosures also has decreased physical abuse at school [Hillier et al., 2010, Writing Themselves in 3. Melbourne: ARCSHS, p. 76], for transgender children and adolescents. Lack of support from parents and particularly fathers was associated with increased physical abuse for transgender and gender diverse youth in the home [Grossman, A. H., D’Augelli, A. R., Howell, T. J., & Hubbard, S. (2005). Parents’ reactions to transgender youths’ gender nonconforming expression and identity;Journal of Gay and Lesbian Social Services, 18(1), 3–16; Jones, T. & Hillier, L. (2013) Comparing Trans-Spectrum and Same-sex- Attracted Youth in Australia: Increased Risks, Increased Activisms, Journal of LGBT Youth, 10:4, 287-307, DOI: 10.1080/19361653.2013.825197]. These correlations should be noted to justify the importance and value of the rest of the study; it shows that parents and guardians’ support is often sought, can be rare in the case of fathers especially, and can have life-saving repercussions. However these findings come from youth studies, which justifies the importance of considering studies directly on parents and guardians themselves more directly… (the aim of the article).

Materials and Methods

The materials and methods section from line 70 on p.2 is a thorough and excellent example of detailed work, that has been well considered. I am unsure why, given this account, a few of the relevant papers above were missed except that it seems the exclusions occurred through the narrow analysis on papers on parents and guardians ‘only’ – we may need that further clarified here.

Results

The author/s need to use common formatting for numbers when written within the text (which is different to how they are written up in tables). For numbers one to nine, words are used when in written text bodies of articles (so for example, the digit ‘9’ on line 144 needs to be written out as ‘nine’. For numbers 10+, digits are to be used (11, 12, 13…). Please go through this section to fix this formatting issue.

Discussion & Conclusion Sections

The discussion and conclusion sections from page 13 of the article is mostly adequate and satisfactory in its representation of the main themes of the papers reviewed. However we do not always have a picture of the ‘point’ of these findings – how do these studies on parents and guardians specifically, compare to the broader literature on transgender and gender diverse youth and children themselves? Does the parent and guardian literature adequately reflect the concerns of transgender and gender diverse youth studies, or contrast strongly on a few points or perspectives around transgender and gender diverse youth (such as their need for support, such as their realisation of their role in protecting them from physical harm in schools and homes or to themselves)? Tease out the meaning of this body of the literature, in relation to transgender and youth studies more broadly, to really bring this piece together more.

Author Response

Review Report Form REVISOR 1

Comments and Suggestions for Authors

Overall this is a strong piece with a few main issues that need to be fixed. There are some wording or numbering muddles, some oversights in the picture given of the established research on parents and guardians from studies of transgender and gender diverse youth and how this study relates to that broader literature, and some extra minor details about the study to be included. A few efforts in these regards to address these issues, will really bring this piece together however, and I see it being a useful piece beyond the parent studies readership and potentially into the much bigger trans youth studies readership if the concerns of trans youth studies are better accounted for and compared here.

Abstract

The abstract is a little clumsy and disconnected from existing literature. For example it opens P.1 Line 1: “Recognizing gender identity in childhood and adolescence by mothers, fathers, or guardians is little explored in scientific literature, nor is emphasis given to the dynamics of secondary social support networks that involves relationship ties with socio-sanitary service institutions in health, education and social fields”. The grammar here is poor as this sentence starts with a continuing verb. Also this sentence relies on assumptions the readers have some pre-existing knowledge of or interest in social support networks (as though this is some widely known and agreed collection of undebated categorisation) and so forth. You need to start an abstract with what is known and you also need to be broader in conceiving who may care for trans youth remembering many are cared for by guardians. For example it is known that “Mothers, fathers, or guardians’ support for disclosures of diverse gender identity has significant relationships with decreased suicidality for transgender children and adolescents”. This is an important and well known association that then makes the lack of exploration of recognition more important. The piece should open with it. Then be a bit clearer on what you mean by social networks (professionals, friends or what groups) – you need to be more direct in defining this, as currently definition occurs too indirectly.

Answer: In the abstract we included an introductory paragraph in accordance with the scientific literature surrounding the theme. Following the revisor's suggestion, we have described the implications of the support from guardians in the decrease of suicide risk amongst gender diverse youth, furthermore, to underpin the central subject of the study, we have also included the necessity of network support so these guardians can support their children.

Introduction

The introduction inherits the somewhat clumsy opening phrasing of the abstract and the strong disconnection from existing literature. For example it opens P.1 Line 31: “Recognizing gender identity by mothers, fathers, or guardians of transgender children and adolescents can occur immediately after disclosure of their own children or in  the course of the search for answers to their desires and individual reactions associated or 33 not with the feeling of “loss” of an idealized child. Again, the grammar here is poor as this sentence starts with a continuing verb, and it would be better to open with “Mothers, fathers, or guardians’ recognition of gender identity can occur…”, or to start the sentence with something like ‘When transgender children…’.  

The introduction like the abstract also needs to start with a stronger justification about why “Mothers, fathers, or guardians’ support for gender identity” matters… why it should be studied, beyond the mere claim that it has not been studied; given it has been studied and seminal correlations uncovered that have informed parental education groups like PFLAG for years.  It is therefore instead important to note from the existing literature that transgender and gender diverse youth are significantly more likely than cisgender same-sex attracted youth to have disclosed their identity to most people in their lives including their parents, and that fathers are comparatively the most likely to reject transgender and gender diverse youth [Jones, T. & Hillier, L. (2013) Comparing Trans-Spectrum and Same-sex- Attracted Youth in Australia: Increased Risks, Increased Activisms, Journal of LGBT Youth, 10:4, 287-307, DOI: 10.1080/19361653.2013.825197]. Again, it is most directly notable in the existing literature that “Mothers, fathers, or guardians’ support for gender identity disclosures” has highly significant relationships with decreased suicidality [Hillier et al., 2010, Writing Themselves in 3. Melbourne: ARCSHS, pp. 74-75], for transgender children and adolescents.  Mothers, fathers, or guardians’ support for gender identity disclosures also has decreased physical abuse at school [Hillier et al., 2010, Writing Themselves in 3. Melbourne: ARCSHS, p. 76], for transgender children and adolescents. Lack of support from parents and particularly fathers was associated with increased physical abuse for transgender and gender diverse youth in the home [Grossman, A. H., D’Augelli, A. R., Howell, T. J., & Hubbard, S. (2005). Parents’ reactions to transgender youths’ gender nonconforming expression and identity;Journal of Gay and Lesbian Social Services, 18(1), 3–16; Jones, T. & Hillier, L. (2013) Comparing Trans-Spectrum and Same-sex- Attracted Youth in Australia: Increased Risks, Increased Activisms, Journal of LGBT Youth, 10:4, 287-307, DOI: 10.1080/19361653.2013.825197]. These correlations should be noted to justify the importance and value of the rest of the study; it shows that parents and guardians’ support is often sought, can be rare in the case of fathers especially, and can have life-saving repercussions. However these findings come from youth studies, which justifies the importance of considering studies directly on parents and guardians themselves more directly… (the aim of the article).

Answer:

All requests have been met. Following the suggestions, we reformulated the introduction to present an overview of the lack of support from those responsible for trans children. Addressing exclusion and family violence is important to contextualize what the scientific literature brings to the table on the subject. At the end of the introduction, we highlight the importance of listening to the guardians in order to contemplate their perspective.

Materials and Methods

The materials and methods section from line 70 on p.2 is a thorough and excellent example of detailed work, that has been well considered. I am unsure why, given this account, a few of the relevant papers above were missed except that it seems the exclusions occurred through the narrow analysis on papers on parents and guardians ‘only’ – we may need that further clarified here.

Answer:

The method followed the systematic review steps with methodological rigor at all stages, in accordance with Joanna Briggs Institute Manual for Evidence Synthesis - Systematic.

Results

The author/s need to use common formatting for numbers when written within the text (which is different to how they are written up in tables). For numbers one to nine, words are used when in written text bodies of articles (so for example, the digit ‘9’ on line 144 needs to be written out as ‘nine’. For numbers 10+, digits are to be used (11, 12, 13…). Please go through this section to fix this formatting issue.

Answer: revised

Discussion & Conclusion Sections

The discussion and conclusion sections from page 13 of the article is mostly adequate and satisfactory in its representation of the main themes of the papers reviewed. However we do not always have a picture of the ‘point’ of these findings – how do these studies on parents and guardians specifically, compare to the broader literature on transgender and gender diverse youth and children themselves? Does the parent and guardian literature adequately reflect the concerns of transgender and gender diverse youth studies, or contrast strongly on a few points or perspectives around transgender and gender diverse youth (such as their need for support, such as their realisation of their role in protecting them from physical harm in schools and homes or to themselves)? Tease out the meaning of this body of the literature, in relation to transgender and youth studies more broadly, to really bring this piece together more.

Answer: We have included a paragraph linking the studies carried out with trans people with those carried out with guardians. We bave also pointed out the importance of listening to both parents and children to achieve a more consistent understanding of the subject.

Reviewer 2 Report

1.  Line 56.  This reviewer is not clear on what "projectal" means.

2.  Line 74.  If the paper referenced has been accepted/published, that should be recognized.

3.  Figure 1.  Depending on the journal's style, numbers in the thousands should have a comma rather than a period. 

4.  Line 166  I'd suggest "studies" rather than "studie,s"

5.  Perhaps the theory here is not complex enough.  Transgenderism has been associated with a number of wild cards, such as being gay but not wanting to be gay, having experienced sexual trauma previously that has led to confusion, bullying by  other children or adults.  If one "accepts" transgenderism too quickly without considering such other factors, then you might be doing a child a disservice.  I also think that some transgenderism may result from social prejudice against normal variations in gender expression which leads to overcompensation.  As in, "they" think I am a sissy, so I will just become a girl to spite them or because I have overinternalized their criticisms. 

6.  This is a personal beef but I continue to fail to understand how we can be paying school administrators huge salaries and yet they don't seem to be able to eliminate bullying against transgender children.  Instead they often marginalize the T/G child!

Author Response

1. Line 56.  This reviewer is not clear on what "projectal" means.

Answer: Revised

2. Line 74.  If the paper referenced has been accepted/published, that should be recognized.

Answer: Revised

3. Figure 1.  Depending on the journal's style, numbers in the thousands should have a comma rather than a period.

Answer: Revised

4. Line 166  I'd suggest "studies" rather than "studie,s"

Answer: Revised

5. Perhaps the theory here is not complex enough.  Transgenderism has been associated with a number of wild cards, such as being gay but not wanting to be gay, having experienced sexual trauma previously that has led to confusion, bullying by  other children or adults.  If one "accepts" transgenderism too quickly without considering such other factors, then you might be doing a child a disservice.  I also think that some transgenderism may result from social prejudice against normal variations in gender expression which leads to overcompensation.  As in, "they" think I am a sissy, so I will just become a girl to spite them or because I have overinternalized their criticisms. 

Answer: Yes. There are many controversies regarding being trans, but we agree with free gender expression from the first years of life, without judgments and instead, more favorable conditions for freedom, respect and consideration for the trans person and their family's demands.

6. This is a personal beef but I continue to fail to understand how we can be paying school administrators huge salaries and yet they don't seem to be able to eliminate bullying against transgender children.  Instead they often marginalize the T/G child!

Answer: Structural transphobia prevents advances in knowledge and ignorance results in bigotry, the access to trans-specific education needs greater reach and support in spaces of power.